# The Indirect Carbon Cost of E-Mobility for Select Countries Based on Grid Energy Mix Using Real-World Data

**Nana Kofi Twum-Duah** [1,*], **Lucas Hajiro Neves Mosquini** [1,2], **Muhammad Salman Shahid** [1], **Seun Osonuga** [1], **Frédéric Wurtz** [1,*] and **Benoit Delinchant** [1]

1   Univ. Grenoble Alpes, CNRS, Grenoble INP*, G2Elab, 38000 Grenoble, France;
    lucas-hajiro.neves-mosquini@grenoble-inp.fr (L.H.N.M.);
    muhammad-salman.shahid@g2elab.grenoble-inp.fr (M.S.S.); seun.osonuga@g2elab.grenoble-inp.fr (S.O.);
    benoit.delinchant@grenoble-inp.fr (B.D.)
2   University of Applied Sciences of Western Switzerland, Energy Institute, HEIA-FR,
    1700 Fribourg, Switzerland
*   Correspondence: nana-kofi-baabu.twum-duah@grenoble-inp.fr (N.K.T.-D.);
    frederic.wurtz@grenoble-inp.fr (F.W.)

**Abstract:** Electric vehicles are considered by many as an emission-free or low-emission solution to meet the challenge of sustainable transportation. However, the operational input, electrical energy, has an associated cost, greenhouse gasses, which results in indirect emissions. Given this knowledge, we pose the following question: "Are zero-emission transportation targets achievable given our current energy mix?" The objective of this article is to assess the impact of a grid's energy mix on the indirect emissions of an electric vehicle. The study considers real-world data, vehicle usage data from an electric vehicle, and carbon intensity data for India, the USA, France, the Netherlands, Brazil, Germany, and Poland. Linear programming-based optimization is used to compute the best charging scenario for each of the given grids and, consequently, the indirect emissions are compared to those of a high-efficiency 1.5 L diesel internal combustion engine for the vehicle: a 2019 Renault Clio dCi 85. The results indicate that for grids with low renewable energy penetration, such as those of Poland and India (Maharashtra), an electric vehicle, even when optimally charged, can be classified as neither a low- nor zero-emission alternative to normal thermal vehicles. Also, for grids with elevated levels of variation in their carbon intensity, there is significant potential to reduce the carbon footprint related to charging an electric vehicle. This article provides a real-world perspective of how an electric vehicle performs in the face of different energy mixes and serves as a precursor to the development of robust indicators for determining the carbon reductions related to the e-mobility transition.

**Keywords:** e-mobility; carbon impact; linear programming; optimization; indirect emissions; electric vehicles; sustainable transportation; grid energy mix

## 1. Introduction

Globally, governments, institutions, and individuals are making the shift towards more sustainable and environmentally friendly practices. Global warming, one of the main proponents of this shift, has necessitated a much-needed change in the conception, design, and implementation of solutions at all levels of society. At the policy scale, the focus has been on the reduction of greenhouse gases (GHGs, which have been identified as one of the key drivers of global warming). Thus, more policies have been conceived and implemented to reduce GHG emissions, especially from major GHG-emitting sectors (energy, transport, agriculture, etc.).

The transportation sector, accounting for 23% of global emissions, ranks high among the main contributors to global warming [1]. According to the European Environment Agency, in

2022 cars (i.e., passenger vehicles) accounted for 60.6% of the carbon emissions related to road transportation (see Figure 1). Thus, decarbonizing the transportation sector can be considered an essential and significant step to achieve our energy transition [2] objectives.

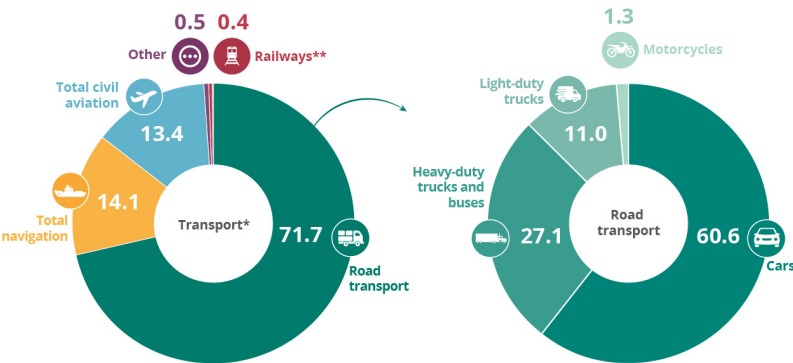

**Figure 1.** Greenhouse gas emissions from transport in the EU in 2022 (source: [3]).

Further, the European Statistics body, Eurostat, has identified the transportation sector as the only sector with GHG emissions higher than those of 1990, notwithstanding the mitigation efforts put in place [4], pertaining to the sector's heavy reliance on petroleum-based products (for fuel, lubrication, etc.) [5]. It is, therefore, not surprising that the sector has seen major reforms following the Paris Agreement [6,7]. The re-emergence of electric vehicles (EVs) [8] is acclaimed as an emission-free alternative to fossil fuel-driven vehicles (i.e., the key to a carbon-free transportation sector). Therefore, it is seen to influence major policy shifts at all levels of society.

In the international landscape, the signatories to COP26 have resolved to achieve 100% emission-free vehicle sales by 2040 [9]. Additionally, many governments (particularly in developed economies) offer bonuses to individuals and businesses who purchase an EV [10–12]. At the commercial level, most vehicle manufacturers have announced a shift to a 100% electric or emission-free product lineup [13], with businesses and organizations also making a shift to 100% electric fleets [14]. At the individual level, the sales figures of electric vehicles show an exponential increase in the sales of EVs [14].

To provide some context, EV sales in the USA recorded a year-on-year increase of 81% in 2018 [15,16] in response to state-level policies across the USA, such as California's Zero-Emissions Vehicle policy [17] which mandates that EVs should have a minimum market share. This is a commendable effort provided there are similar efforts to increase electricity production to match the increased demand from EVs, especially using low-emission or zero-emission energy resources. Given that all the countries selected for this study have committed to a net-zero scenario by 2050, Table 1 shows the change in GHG emissions associated with the energy sector in 2022 relative to the 1990 levels as well as the current EV stock.

**Table 1.** Summary of country efforts towards decarbonizing their respective energy sectors (data sources: [18–21]).

| Country | % Change in GHG Emissions in the Energy Sector Until 2021-22 Relative to 1990 Levels | % of EVs in New Vehicle Sales as of 2023 |
|---------|---------------------------------------------------------------------------------------|------------------------------------------|
| France | −38% | 25% |
| Germany | −42% | 24% |
| Poland | −35% | 6.6% |
| Brazil | +142% | 3% |
| USA | −1.77% | 9.5% |
| India | +288% | 2% |

While EVs by themselves have zero tank-to-wheel emissions (i.e., direct emissions on the road), it is still important to take into consideration that, just like fossil fuel-driven vehicles, EVs require energy to provide their core service (i.e., transporting people and goods from one point to the other). Thus, the energy source is an essential but often ignored parameter (see Figure 2) that impacts how emission-free an EV is.

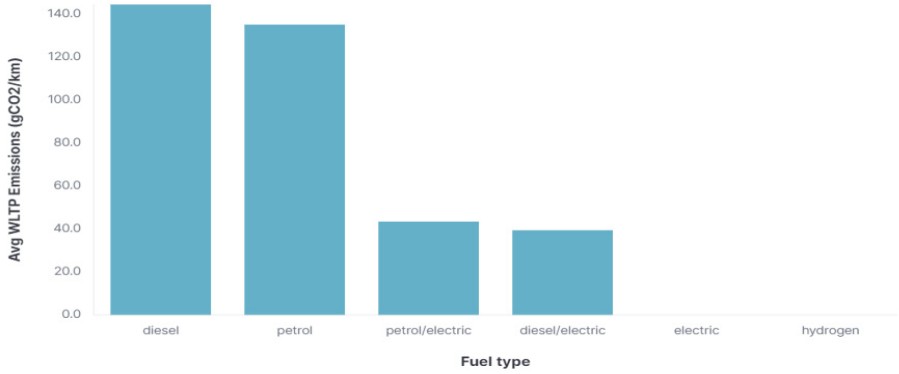

**Figure 2.** Average Worldwide Harmonized Light Vehicles Test (WLTP) emissions of vehicles by fuel type in the European Union (source: [22]).

This article evaluates the impact of a grid's energy mix on an EV's carbon footprint. The research work is based on real-world data from an electric vehicle and the electric grids of various countries. Based on these data, an optimal strategy is computed to evaluate the best-case scenario with the given datasets. Section 2 presents the problem statement, Section 3 presents a literature review, Section 4 provides an overview of the data used for this study, and Section 5 details the optimization approach as our methodology, whereas Section 6 presents the results, Section 7 is a discussion of the results, and Section 8 presents the conclusions.

## 2. Problem Statement

Given the current energy transition goals, i.e., a transition to 100% electric mobility (for light and medium vehicles) by 2040, this article seeks to address the following questions: "Are zero-emission transportation targets achievable given our current energy mix?"; under what conditions can these targets be considered feasible?

We consider real-world data from an EV and various national electric grids (with varying energy mixes) to assess the impact of the energy mix of a country's electric supply on the indirect emissions of the EV.

Additionally, since EVs have been demonstrated as a viable source of indirect flexibility [23], we assess the potential for indirect emission reduction by optimizing the charging of the EV. The contributions of this article are summarized as follows:

An assessment of the grid energy mix of the respective country on the well-to-wheel (WTW) emissions (i.e., indirect emissions) of an electric vehicle using real-world data

A comparison of a best-case scenario (optimal case) with the reference case scenario (the real-world case).

## 3. Literature Review

Contrary to what is advertised by vehicle manufacturers, EVs should not be labeled as "zero emissions", at least not with the state of electricity grids across the globe. This "zero-emissions" assertion is based solely on the consideration of tank-to-wheel (TTW) emissions (i.e., operational emissions). This approach, however, neglects the upstream emissions related to the electricity source (i.e., the wheel-to-tank (WTT) emissions). In other words, in order to accurately represent the emissions of an EV, the well-to-wheel

(WWT) emissions of the electrical energy used to power the EV's drivetrain must be considered, not just the TTW emissions, as is usually the case.

Skrúcaný et al. [24] in their works found that EVs might not always be environmentally friendly. Their results showed a strong correlation between the efficiency of the electricity generation technology and the main primary fuels used within the country. Interestingly, Skrúcaný et al. [24] show that for countries with a high-carbon-intensity electrical grid, such as Poland, an ICE is a more environmentally friendly choice as it has a lower carbon footprint relative to an electric vehicle when the well-to-wheel emissions of both types of cars are compared.

Additionally, the life cycle assessments (LCAs) of electric vehicles show relatively higher embodied energy and consequently higher embodied carbon emissions of electric vehicles [25]. Depending on the carbon intensity of the input electricity, this higher embodied carbon emission can be offset relatively quickly (however, it is dependent on factors such as energy mix and generation efficiency, as pointed out by Skrúcaný et al. [24]). LCAs allow for the evaluation of the environmental impacts of a product during its entire life cycle, from the extraction of raw materials until its end-of-life stages, as defined by ISO14040 [26]. This methodology reduces the possibility of shifting burden, i.e., reducing the environmental impact of an element might increase the environmental impact of another element. While, on the one hand, studies like [27,28] identified a 50% and 44% improvement in overall global warming potential given a transition to EVs, studies like Cellura et al. [29], on the other hand, calculated an average of 500% increase in abiotic resource depletion (such as Lithium and Cobalt [30]) for EVs when compared to internal combustion engine (ICE) vehicles while performing an LCA assessment.

Furthermore, the burden shift phenomenon [31] can also happen between the various stages of the vehicle's life cycle, namely the production, utilization, and recycling stages, as identified by Verma et al. [32]. Indeed, GHG emissions linked to EV production are 127% to 200% greater than an equivalent ICE vehicle, as evidenced by [27,28] (both estimates based on the SuperLightCar European project model [33]). Therefore, together the higher depletion of resources and embodied $CO_2$ implies that at the manufacturing phase, EVs score lower in terms of global warming potential (GWP) relative to ICE vehicles. Further compounding this issue is the ever-growing demand for EVs to have longer range, which given the current storage technology constraints means bigger batteries (and consequently heavier vehicles), in reality (to provide sustainable alternatives to urban mobility challenges). Perhaps, in the near future, EV designers might utilize and benefit from innovation concepts such as Low-Tech Innovation [34] and Eco-Design [35].

On the other hand, A 100% EV future would also imply technical challenges on the national grids. These challenges include but are not limited to power losses, voltage instability, etc. for electricity network operators [36]. Furthermore, there is also the issue of meeting the increased demand for energy; National Grid estimates that by 2030, EV charging stations along highways in the USA will require a connection capacity of over 5MW [35]. Hussain et al. [36] review multiple strategies (coordination, EV charging functioning, fleet sharing, etc.) for managing the envisioned technical constraints. The smart grid seeks to address most of these challenges; however, it is also imperative that in trying to solve one problem we do not create another (which might be even more difficult to solve).

Hybrid Electric Vehicles (HEVs) and Plug-in Hybrid Electric Vehicles (PHEVs) are alternative solutions that exist and allow for the transition toward an ecologically viable and sustainable future transportation landscape [37]. To illustrate this point, a study in Indonesia compared EVs, PHEVs, HEVs, and ICEs and found that whilst EVs had the lowest $CO_2$ emissions, EVs contributed the most towards Nitrogen oxide (NOx) and nitrous oxide (N2O) emissions (however, this was country-specific and highly dependent on the energy mix) [38]. Veza et al. [38] also showed that HEVs' and PHEVs' $CO_2$ emissions were significantly reduced compared to those of conventional ICEs and that HEVs offer the best balance between the factors of comparison considered in their work (i.e., selling price, GHG emission cost, and maintenance cost).

## 4. Data Analysis

For this study, two datasets, the EV and grid carbon intensity datasets, are considered for the period between January 2020 and December 2020 (1 year). The data used are at a one-hour time step and were preprocessed to remove outliers. The authors of this article subscribe to the principles of Open Science. As such, the data, notebooks, and code related to this study are available online and follow the Open Reproducible Use Case for Energy (ORUCE) principles [39].

### 4.1. EV Dataset

This dataset (the dataset was obtained as part of an ongoing project to collect data from electric vehicles; more information can be found here: EVE project (FR)) was extracted from a 2013 Renault ZOE with a 22 kWh battery [40]. The vehicle was used as a personal vehicle in southeastern France. Figure 3 shows the daily statistics related to the usage of the EV. Charging usually occurred with a power of 7 kW (per hour) or less (see Figure 3d below) and the vehicle was frequently used to cover a daily distance of 40 km or less, as depicted in Figure 3f below. The charging data acquired from the EV were measured after losses; thus, the energy from the grid was calculated as a ratio of the measured charging power to the charging efficiency.

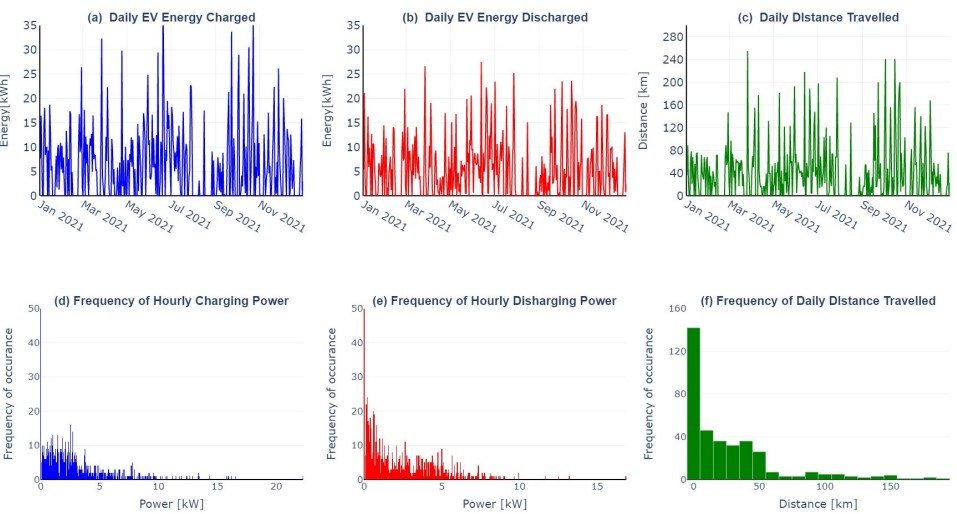

**Figure 3.** Daily usage summary of EV: (**a**) energy charged, (**b**) energy discharged, and (**c**) distance traveled. Distribution of EV data: (**d**) frequency of power at which EV was charged, (**e**) frequency of power at which EV was discharged and, (**f**) frequency of hourly distance covered by EV (data source: [40]).

Additionally, to ensure that the EV data were representative of an average driver, the vehicle speed over the evaluation period was analyzed (see Figure 4). Also, Figures 4a and b show that driving usually occurred at 30 and 50 km/h. However, from Figure 4a it can be observed that higher speeds of up to 80 km/h were recorded, particularly in June and July. Moreover, Figure 4b indicates that the vehicle was frequently used to commute less than 40 km per day. Thus, it can be inferred from Figure 4 that the vehicle conforms to average European personal vehicle usage [41] and is typically used for short-distance commutes (100 km or less [41,42]).

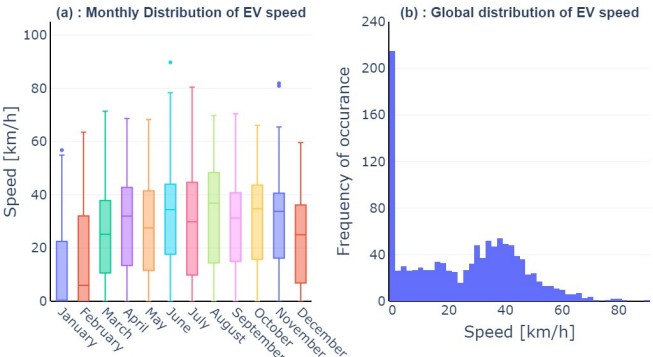

**Figure 4.** Statistical distribution of vehicle speed: (**a**) monthly distribution and (**b**) annual distribution (data source: [40]).

### 4.2. Grid Carbon Intensity Dataset

Electricity, as previously mentioned, also has associated GHG emissions, not only from production but also for transportation and distribution (infrastructure and losses). Thus, the carbon intensity of a grid's electricity will be highly dependent on the mix of production at a given time. Additionally, despite having a much lower carbon impact than coal and gas, renewable energies do indeed emit GHGs.

For this study, we considered electricity carbon intensity (well-to-wheel) data from Electricitymap [43]. Table 2 shows a summary of the dataset (the methodology used for estimating the carbon emissions in this dataset is detailed here: electricitymaps.com) used for this study.

**Table 2.** Summary of the grid carbon intensity data for 2020 (calculated by the authors based on the dataset obtained from [43]).

| Country | Initials | Mean Emissions [gCO$_2$e/kWh] | Max Emissions [gCO$_2$e/kWh] | Min Emissions [gCO$_2$e/kWh] | Variance | Major Sources |
|---|---|---|---|---|---|---|
| France | FR | 59.03 | 132.6 | 21.73 | 653.24 | Nuclear Gas Hydro |
| Germany | DE | 329.67 | 513.13 | 108.48 | 8193.33 | Coal Gas Wind |
| Netherlands | NL | 363.20 | 509.91 | 45.68 | 7078.21 | Gas Coal Biomass |
| Poland | PL | 638.50 | 767.12 | 404.65 | 4089.20 | Coal Wind Gas |
| Brazil (North) | BR-N | 183.33 | 355.40 | 49.35 | 6303.19 | Hydro Wind |
| India (Maharashtra) | IN-MH | 728.14 | 757.41 | 588.06 | 779.32 | Coal |
| USA (California) | US-CAL-CISO | 260.74 | 387.45 | 98.43 | 3538.42 | Solar Gas Wind |

Similarly, Figure 5 shows the hourly carbon intensity of the various grids considered for the period January to December 2021. From the presented data, it is important to highlight the staggering contrast between these seven electricity mixes when comparing the average values and amplitudes of the observed data. India (Maharashtra), Figure 5b, is seen to have the most carbon-intense grid (largely fueled by coal), while France, which has a high use of nuclear energy, is seen to have the grid with the lowest CI. Equally interesting are the German, Dutch, and Californian (USA) grids as they have a high proportion of renewable energy resources in their energy mix.

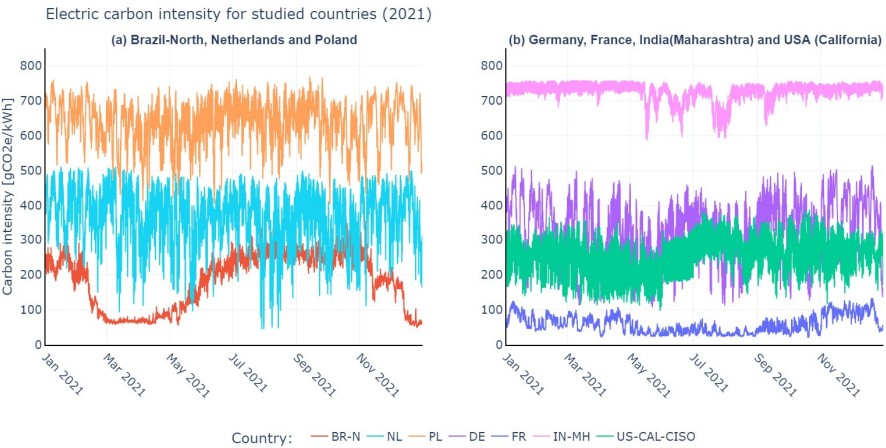

**Figure 5.** Hourly carbon intensity for the electricity grids of (**a**) Northern Brazil, the Netherlands, Poland, (**b**) Germany, France, India (Maharashtra), and the USA (California) (data source: [43]).

To further investigate carbon intensity variations, a Fast Fourier Transform (FFT) was applied to the data for France, the Netherlands, California, and India, and the results are shown in Figure 6 below. From Figure 6, the frequency domain was converted to the period in hours to improve readability. Indeed, in all investigated countries, the daily variation is a key component of the spectrum, represented by the peaks in magnitude for the 24, 12, and 6 h periods. Additionally, peaks at 168 and 84 h occur from the weekly variation in carbon intensity; however, these are more visible in the European nations, where industrial and commercial activities are much slower on weekends (particularly on Sundays) compared to weekdays. The production profile has a similar spectrum with the same peaks as the one identified above, as evidenced by Roux et al. [44].

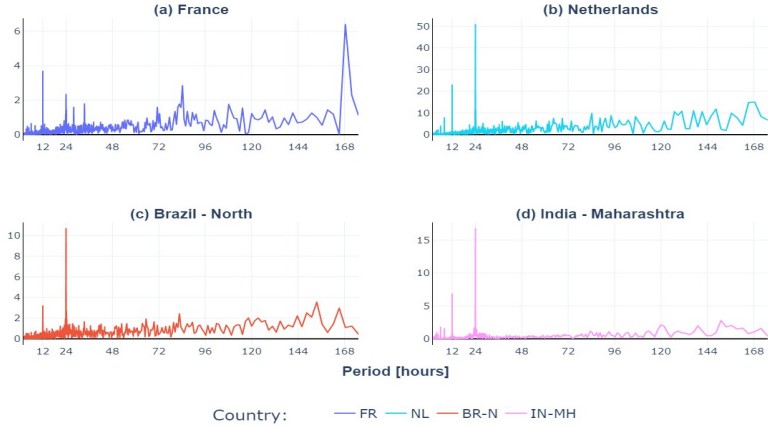

**Figure 6.** Results of the Fast Fourier Transform applied to carbon intensity in hours for (**a**) France, (**b**) the Netherlands, (**c**) California (USA), and (**d**) India (data source: [43]).

This amplitude of carbon intensity and the fundamental periods in which it varies, notably daily and weekly, presents an opportunity for the optimizer (discussed in Section 5) to shift EV charging not only to different hours but also to different days and weeks.

Given the characteristics of the data, particularly the grid carbon intensity, it should be possible to determine an optimal charging schedule that minimizes the indirect carbon associated with the charging of the EV. The subsequent section presents said optimal strategies.

## 5. Methodology: Optimal EV Charging

For this article, our objective is to use real-world data to assess the carbon impact of EVs in different energy mixes. To achieve this objective, we first assessed the emissions of our subject EV in the chosen countries. Subsequently, we computed the best case based on the charging and discharging (i.e., driving) behavior obtained from our EV dataset.

To compute the best-case scenario, we considered a linear programming (LP) approach to optimize the charging of the EV. LP is one of many mathematical approaches used for solving optimization problems (maximizing or minimizing a variable). LP formulations are simple but can be applied to complex problems, and have been used to solve complex industrial, military, and financial problems [45]. LP has been demonstrated to be effective and efficient in solving energy-related optimizations with minimal computational cost. Additionally, the linear constraints result in a convex feasible region, ensuring a global optimum (provided the problem is well defined) [46,47]. Figure 7 depicts the system used for the optimization and uses the OMEGAlpes graphical representation [48].

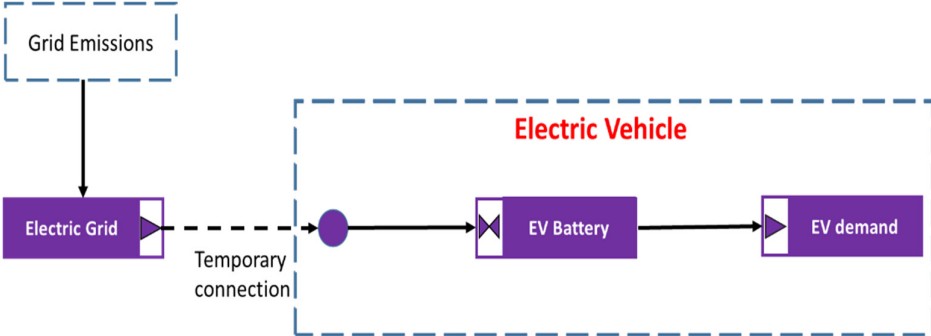

**Figure 7.** Graphical representation of the energy system for optimization.

The objective function for the optimization considered was to minimize the carbon impact related to the charging of the EV given the data available. Two optimization strategies were considered for this article and are detailed below.

Day strategy: For this strategy, a horizon of one year was considered; however, an additional constraint was added in the form of a sub-horizon (one day). The goal was to respect the charging demand associated with the charging of the vehicle. Thus, for each day, the same quantity of energy was charged, as was the case in the dataset; however, the goal of the new charging schedule was to reduce the carbon impact of EV charging. The discharging of the EV was respected, implying that charging could not happen during a time step for which discharging originally happened (i.e., the vehicle was being driven). This approach considered (to some extent) the driver's behavior and proposed an optimal strategy.

Annual Strategy: For this strategy, the daily charging constraints were not considered. Thus, it is possible to displace charging across different days provided there is sufficient energy in the EV's battery to meet the driver's needs. The optimization was conducted for the entire horizon (one year) and was expected to change the charging schedule of the EV. The discharge schedule (as originally provided in the EV dataset) was respected. Thus, the original driving (which directly implies the discharging schedule) of

the EV was considered, ensuring that this strategy would still ensure that the transportation needs of the EV owner were not violated (i.e., there is always enough energy stored in the battery to satisfy the energy demand of the EV as it was used in real life).

Given the two strategies defined above, the objective function of the optimization was thus defined mathematically as follows [49]:

$$objective = \left[ \sum_{pd=0}^{pd_{max}} \sum_{t=0}^{23} p_{charge}(pd, t) \times emissions_{grid}(pd, t) \right] \tag{1}$$

where $P_{charge}(pd, t)$ is the charging power from the grid and $CO2_{grid}(pd, t)$ is the grid carbon intensity for the sub-horizon pd (one day, in the case of the day strategy, and one year for the annual strategy) at time step t. Moreover, to ensure that the battery state of charge (SOC) stays within defined operational bounds,

$$SOC_{Min} \times Cap_{bat} \leq E_{bat}(pd, t) \leq SOC_{Max} \times Cap_{bat} \tag{2}$$

where $SOC_{Min} \times Cap_{bat}$ and $SOC_{Max} \times Cap_{bat}$ refer to the minimum and maximum bounds of battery energy, respectively, and $E_{bat}(pd, t)$ is the electric charge in the battery.

The energy in the battery $E_{bat}(pd, t)$ is thus also defined as follows:

$$E_{bat}(pd, t) = E_{bat}(pd, t-1) + \left[ P_{charge}(pd, t) \times \eta_{charge} - \frac{P_{discharge}(pd, t)}{\eta_{discharge}} \right] \times ts \tag{3}$$

$P_{discharge}(pd, t)$ is the discharge power of the battery for the sub-horizon pd at time t, $\eta_{charge}$ and $\eta_{discharge}$ are the battery charge and discharge efficiencies, respectively, and $ts$ is the timestep coefficient (defined as the ratio of the time step in minutes to 60 min). In addition, to ensure charging and discharging respect the technical constraints of the battery and the vehicle movement,

$$P_{charge}(pd, t) \leq P_{max-charge} \times availability(pd, t) \tag{4}$$

$$P_{discharge}(pd, t) \leq P_{max-discharge} \times availability(pd, t) \tag{5}$$

where $P_{max-charge}$ and $P_{max-discharge}$ are the maximum charging and discharging power of the EV's battery and $availability(pd, t)$ is a binary value which is determined by the discharge power of the EV (i.e., it has a value of one when the vehicle is not being discharged and zero when the vehicle is in motion). To ensure an energy balance in the system,

$$P_{grid}(pd, t) - P_{charge}(pd, t) + P_{discharge}(pd, t) - P_{demand}(pd, t) = 0 \tag{6}$$

where $P_{grid}(pd, t)$ and $P_{demand}(pd, t)$ are the power imported from the grid and consumed by the EV, respectively. Lastly, to ensure continuity in the battery's state of charge (SOC), particularly in the day strategy, an additional constraint was added to ensure that the battery SOC stayed within the defined operating bounds.

$$E_{bat}(pd + 1, 0) = E_{bat}(pd, T+1) + \left[ P_{charge}(pd, T) \times \eta_{charge} - \frac{P_{discharge}(pd, T)}{\eta_{discharge}} \right] \times ts \tag{7}$$

where $T$ is the last time step in the set defined by the sub-horizon given by $\{0, 1, 2 \dots T\}$, and the final battery energy is constrained as defined in Equation (2). Thus, the starting battery energy for the various periods was defined as follows:

$$E_{bat}(pd, t) = \begin{cases} E_{bat}(pd, T) \leq Cap_{bat}, & if\ pd = 0\ and\ t = 0 \\ E_{bat}(pd-1, T+1), & if\ pd > 0\ and\ t = 0 \end{cases} \tag{8}$$

In order to carry out the optimization, the following technical parameters outlined in Table 3 were considered. Both day and annual strategies were modeled as PYOMO [50] concrete models and were solved using the Gurobi solver (version 9.1.2) [51].

**Table 3.** Technical parameters and assumptions considered for the optimization.

| Parameter | Unit | Value |
|---|---|---|
| Max charging power | kW | 20.0 |
| Max discharging power | kW | 40.0 |
| Charging efficiency | % | 85.0 |
| Discharging efficiency | % | 100.0 |
| Sub-horizon (day strategy) | days | 1 |
| Sub-horizon (annual strategy) | days | 365 |
| Horizon | days | 365 |

The optimization assumes a constant charging efficiency; however, in reality, different parameters would affect the efficiency value (charging power, single or three phase). However, because this error is carried over into all optimizations, it should not impact the results and conclusions of this study.

## 6. Results

### 6.1. Carbon Footprint of the EV

To assess the indirect emissions of the EV, we first compute the aggregated monthly carbon footprint of the EV (Figure 8). Essentially, Figure 8 depicts the product of the hourly charging energy and the hourly average CI (depicted here in $gCO_2e$), and is given mathematically as follows:

$$Emissions_{EV} = \sum_{t=0}^{t_{max}} P_{charge}(t) \times Emissions_{grid}(t) \qquad (9)$$

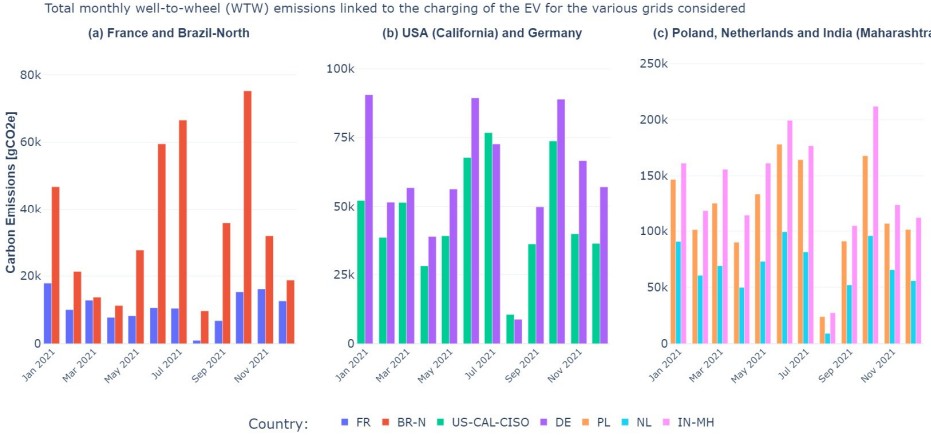

**Figure 8.** Total monthly WTW (indirect) emissions resulting from the charging of the EV (derived from historical data) for (**a**) France and Northern Brazil, (**b**) the USA (California) and Germany, and (**c**) Poland, the Netherlands, and India (Maharashtra).

### 6.2. Estimated Indirect Emissions Based on Grid Energy Mix

Based on the estimated monthly carbon footprint shown in Section 6.1, the monthly WTW emissions were estimated by considering the monthly distance traveled (see Figure 9). Thus, the average monthly WTW emissions can be represented as follows:

$$Emissions_{indirect} = \frac{Emissions_{EV}}{\sum distance(t)} \qquad (10)$$

The average well-to-wheel emissions of a similarly sized (and comparable vehicle class) 2019 Renault Clio dCi 85 with a 1.5L diesel engine (detailed specification of the vehicle can be found here: https://www.ultimatespecs.com/ accessed on 20 June 2024) and having a worldwide harmonized light vehicle test procedure (WLTP, i.e. the tank-to-wheel) emissions of 110 gCO2/km [52] was used as a reference ICE vehicle for comparison. Considering that the WLTP emissions values are tank-to-wheel emissions, and cannot be fairly compared to well-to-wheel values (as have been calculated), we estimated the well-to-tank values as follows:

$$Emission_{WTW} = Emission_{TTW} + Emission_{WTT} \tag{11}$$

where $Emission_{WTW}$, $Emission_{TTW}$ and $Emission_{WTT}$ represent well-to-wheel, tank-to-wheel, and well-to-tank emissions, respectively. From the literature, the well-to-tank emissions for diesel typically range from 15 to 20 gCO2e/MJ; we consider an average of 18 gCO2e/MJ [53]. Thus, considering the energy density of diesel, 137 MJ/l, and the average fuel consumption of the reference ICE (4.2l/100 km[3]), we estimated an average well-to-tank emission as follows:

$$Emission_{TTW} = \frac{18\ gCO_2eq\ /MJ \times\ 137\ MJ/l}{23.8 km/l} = 103.572 gCO_2e/km \tag{12}$$

Thus, $Emission_{WTW}$ for the reference ICE vehicle was estimated to be 213.572 gCO2e/km.

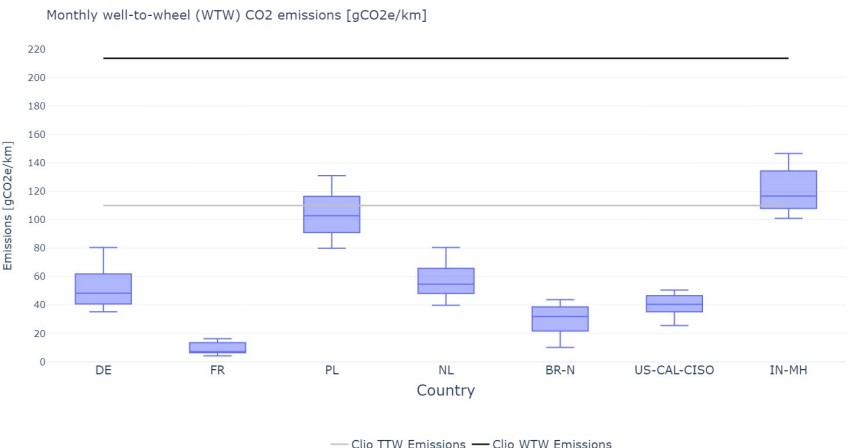

**Figure 9.** Variation in the average monthly WTW (gCO2e/km) emissions in the selected countries.

Additionally, Figure 10 compares the cumulative equivalent carbon emissions of the EV and the reference ICE. We considered the embodied carbon from [28] which are not for the vehicle under consideration (i.e. the Renault Clio or ZOE), but are however, representative of the embodied carbon emissions of a similar-sized EV and ICE at the manufacturing phase. The results below assumed the same charging behavior and grid conditions for 15 years (approximately 150,000km).

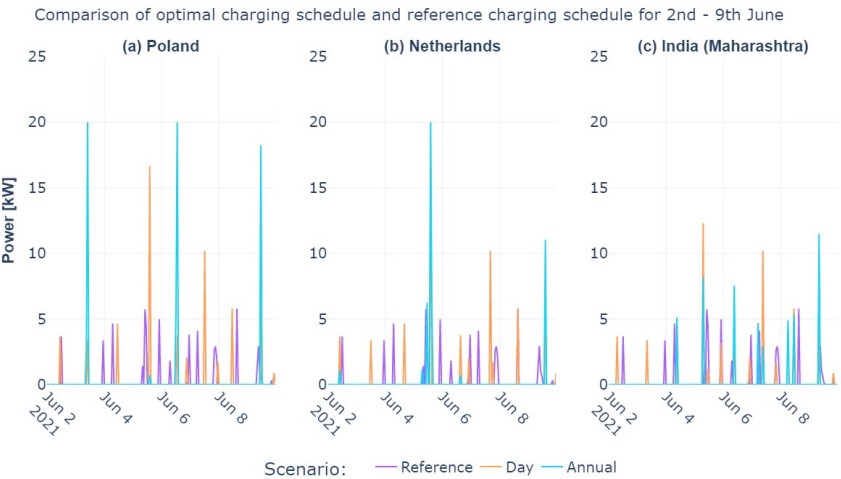

**Figure 10.** Cumulative vehicle life cycle CO₂ emissions for the different studied grids.

*6.3. Optimal Charging Schedule*

Based on the optimization strategies described in Section 5, two new charging schedules were obtained. Figure 11 shows a comparison of the original charging schedule (i.e., the reference case) and the proposed optimal charging schedules (based on the day and annual strategies) for Poland, the Netherlands, and India (Maharashtra); the countries with the worst indirect equivalent emissions as seen in Section 6.1.

**Figure 11.** Comparison of one week's optimal and reference charging schedules for (**a**) Poland, (**b**) the Netherlands, and (**c**) India (Maharashtra).

To further highlight the gains of the optimal charging schedules, the monthly WTW emissions were calculated as described in Section 6.1 for the proposed optimal schedules of Poland, the Netherlands, and India (Maharashtra).

Figure 12 shows a comparison of the reference monthly WTW emissions against those of the optimal strategies. As was the case with Figure 8, the average emissions of the reference ICE vehicle (Renault Clio) were used as a benchmark to evaluate the emissions performance. Additionally, a summary of the results obtained after optimization is detailed in Table 4 below.

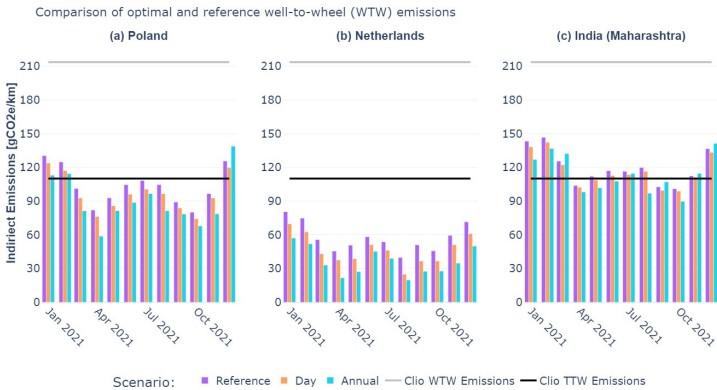

**Figure 12.** Comparison of monthly reference indirect EV emissions against optimal WTW emissions for (**a**) Poland, (**b**) the Netherlands, and (**c**) India (Maharashtra).

**Table 4.** Comparison of optimal strategies with the reference case.

| Country | Carbon Intensity [g CO₂e/kWh] | | | Gain over Reference Case [%] | |
|---|---|---|---|---|---|
| | Reference Case | Day Strategy | Annual Strategy | Day Strategy | Annual Strategy |
| **BR-N** | 416,255.51 | 378,926.18 | 354,428.97 | 8.97 | 14.85 |
| **DE** | 717,362.63 | 603,858.34 | 488,884.49 | 15.82 | 31.85 |
| **FR** | 128,335.11 | 105,583.74 | 88,872.43 | 17.73 | 30.75 |
| **PL** | 140,9601.38 | 1,315,595.33 | 1,217,051.62 | 6.67 | 13.66 |
| **NL** | 792,456.38 | 652,101.03 | 507,538.03 | 17.71 | 35.95 |
| **IN-MH** | 1,643,089.02 | 159,8821.02 | 1,566,472.61 | 2.69 | 4.66 |
| **US-CAL-CISO** | 541,795.62 | 427,855.90 | 377,041.72 | 21.03 | 30.41 |

## 7. Discussion

From Figure 8, which shows the total monthly WTW emissions of the EV in the various grids, France's low carbon energy mix plays to the country's advantage, allowing for the EV to have the lowest carbon impact compared to the other countries studied. California and Germany presented particular interests for the optimization due to the high variability in their grids' carbon impacts. The Indian (Maharashtra) grid, on the other hand, accounted for the highest carbon footprint, this can be attributed largely to the low levels of variations coupled with the high carbon intensity of the energy mix.

The question then arises, "Does an EV hold any real value in terms of carbon emission?" Given the results in Section 7, the influence of the energy mix is a major determining factor that should drive the decision to pursue e-mobility. Whilst the average annual grid CI provides some indication of the potential carbon reduction in e-mobility, it is not sufficient to determine if an EV would be less carbon-intensive compared to an efficient ICE. To illustrate this point, Germany and the Netherlands had average carbon intensities of 330 and 360 g CO₂e/kWh for the evaluation period, respectively. However, given Germany's high Solar PV penetration and consequently the high variance in the CI, the German grid yielded significantly lower WTW emissions compared to those of the Dutch grid (see Figure 9b,c), particularly in the summer months. For the Polish and Indian grids, Figure 9c shows that the Renault Clio would have been the better choice considering "tailpipe" emissions. Consequently, a higher EV penetration could potentially lead to grids with higher carbon intensities (i.e., burden shift); however, the potential carbon reduction related to the reduced production of fuels (petrol and diesel) is anticipated to be significantly higher in comparison [54].

To study the balance between embodied and operational emissions, Figure 10 shows the cumulative emission of greenhouse gasses (in CO₂ equivalent) throughout the vehicle's life cycle. The markings in red highlight the intersection between the cumulative

WTW emissions of the Renault ZOE in different grids and that of the reference ICE vehicle (i.e., the Renault Clio). Thus, these intersections represent the point at which the EV's life cycle emissions equalize those of the ICE's, or how many kilometers the studied EV would need to travel given the grid's carbon intensity to pay back its extra 50–100% embodied carbon emissions.

To provide context, the EV would have to be driven 40,900 km in the French grid to offset the extra embodied $CO_2$ for the TTW scenario and approximately 19,600 km in the WTW scenario.

However, as mentioned in the introduction of this work, life cycle assessments of electric vehicles highlight the importance of the fabrication and assembly processes, notably of the storage systems. Indeed, [28] calculated around 4.8 tCO$_2$e compared to 8.8 tCO$_2$e of embodied emissions from an internal combustion engine and an electric vehicle, respectively (values considered for the embodied $CO_2$ in Figure 10). Also depicted in Figure 10 are the cumulative TTW emissions of the reference vehicle, which when compared to the WTW values show a gross underestimation when only TTW values are considered. For low-carbon-intense grids, such as that of France, it was observed that the ZOE would have paid back the extra embodied carbon at approximately 40,900 km for the TTW-only case and 19,600 km for the WTW case. Similarly, for high-emission grids such as that of India, the ICE curve does not intersect with the EV curve (in fact, the gap widens with increasing distance) for the TTW case, but a payback distance of 41,480 km was achieved for the WTW case. Figure 10, thus, further illustrates the significance of low-carbon-intensity electric grids for the transition to e-mobility.

As highlighted in Section 5, two optimization strategies, the day and annual strategies, were considered. Given that the day strategy constrained the optimization to respect the daily charging requirements from the reference case (i.e., charge as much as was originally charged for each day), it was expected that the results would be less optimal relative to the annual strategy. Figure 11 illustrates how the added flexibility of being able to determine a more optimal charging schedule affected the charging schedule. Also, of equal importance is the reduction in the charging frequency and the increase in the charging power. This phenomenon indicates that the EV was originally charged too frequently and possibly with a low-power charger (3–7 kW rated power).

For India and Poland, an optimal charging schedule did not change the initial results. However, for Poland, there was a noticeable improvement in the WTW emissions, owing to the medium–high variance in the grid's CI. For India on the other hand, the improvements were very minimal and are indicative of the low levels of variation in India's CI data. The Netherlands, on the other hand, showed significant gains, especially considering the annual strategy (17.71% for the day strategy and 35.95% for the annual strategy). For each month, the WTW emissions of the annual optimal strategy proved to be lower than the TTW emissions of the reference ICE vehicle (Renault Clio—110 g CO$_2$e/km [3]). This gain can be attributed in part to the high variance in the Dutch grid. In effect, the variance in a grid's CI is consequently a major parameter that determines how low-carbon an EV would be. Additionally, from Table 4, Germany (16–32%), France (18–31%), and California (20–30%) also showed good prospects for decreasing the carbon footprint of the EV. For France, which was found to have the lowest WTW emissions, this gain implied a reduction in WTW emissions from 15.19 g CO$_2$e/km to 12.50 g CO$_2$e/km for the daily strategy and 10.52 g CO$_2$e/km in the case of the annual strategy.

## 8. Conclusions

The transportation sector is a key carbon-intense sector in achieving our energy transition goal and, more importantly, achieving a sustainable and eco-friendly future. Electric vehicles are one of many (possibly the most popular) low-carbon solutions being developed as a more sustainable alternative to ICE vehicles. Whilst this effort is laudable, this article shows that EVs are not zero-emission options but are a low-carbon alternative to conventional ICE vehicles.

Further, even though studies show that EVs require up to 500% more raw materials (and as such contribute more to abiotic resource depletion) and have higher embodied $CO_2$ (approximately 1.5–2 times) compared to an ICE of similar characteristics, EVs still are the better ecological choice, especially when one considers well-to-wheel emissions.

The energy mix and consequently the carbon impact of a grid is essential in determining if e-mobility is the way to go; however, the assessment needs to be made on WTW emissions and not TTW emissions. As demonstrated with the Indian grid (which had a high CI with low levels of variation), an ICE with higher efficiency would seem a reasonable choice, especially for the immediate future, if one only considers TTW emissions.

Despite our findings showing that EVs are the obvious choice even in the most carbon-intense grids (such as Poland and India (Maharashtra)), the advocacy for EVs must consider the resilience and capacity of the grid to handle the impending increase in demand associated with high EV adoption. For such economies, PHEVs and HEVs would seem to be a better choice as they reduce emissions, are cheaper than EVs, and facilitate the transition towards fully electrified mobility in such countries.

Given the potential uptake of EVs in the near future, measures have to be taken in countries to encourage the optimal charging of EVs. As the carbon intensity varies not only on a daily but also on a seasonal scale, considerations to these measures need to be made for the behaviors to be encouraged on a country-by-country basis. Encouraging optimization on a seasonal scale might reduce the overall impact of emissions and consequently result in a higher level of grid fortifications based on requirements in the summer months.

The research presented in this article is, however, limited to the evaluation of the WTW emissions of EVs given different energy mixes. We do not consider PHEVs and HEVs; however, such a study is envisaged as a future work.

Further, work is needed to develop a robust indicator that considers parameters such as the variation in a grid's carbon intensity and provides an estimate of how low-carbon an EV would be in such a grid. Further, the analyses presented here are based on a seemingly representative personal EV user's behavior. Further work can be carried out to include different use patterns including public transport operations, taxi services, commercial activities, and goods logistics services. Regional variabilities in usage patterns can also be included in future analyses. Finally, modern vehicles provide comprehensive amounts of data, which could provide more insights and inform decision-making at individual and policy levels. It is therefore imperative that more effort is put in place to collect such data and make them available.

**Author Contributions:** All authors contributed to the conceptualization and methodology used in this review. S.O. and N.K.T.-D. contributed towards data acquisition and preparation; N.K.T.-D., M.S.S., L.H.N.M. and S.O. analyzed the literature, modeled the case study, and drafted the article. B.D. and F.W. supervised and directed the research and provided a formal analysis of the review. All authors have read and agreed to the published version of the manuscript.

**Funding:** This work was partially supported by the ANR project ANR-15-IDEX-02. eco-SESA program (https://ecosesa.univ-grenoble-alpes.fr/ accessed on 1 June 2024), Observatory of the Energy Transition— "Observatoire de la Transition Energétique" (https://ote.univ-grenoble-alpes.fr/ accessed on 1 June 2024), and the program Datadistrict supported by the Carnot "Energie du Futur".

**Data Availability Statement:** The original electric vehicle data presented in the study are openly available in Recherche Data Gouv at data.gouv.fr. The data can be cited using the following citation: "Osonuga, Seun; Wurtz, Frederic; Delinchant, Benoit, 2023, "The EVE Pilot: Usage Data from an Electric Car in France", https://doi.org/10.57745/5O6QIH, Recherche Data Gouv, V2, UNF:6:FxFjLc9W3RRzI7VHLe8bIQ== [fileUNF]." The original carbon intensity data for the various grids studied can be directly obtained from electricity maps (electricitymaps.com). The data and associated notebooks used in this study are available in the following repository: https://gricad-gitlab.univ-grenoblealpes.fr/NanaKofi/ev_study.

**Conflicts of Interest:** The authors declare no conflict of interest.

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
