# Peer review of "The Indirect Carbon Cost of E-Mobility for Select Countries Based on Grid Energy Mix Using Real-World Data"

_sustainability, doi:10.3390/su16145883_

Round 1

Reviewer 1 Report

Comments and Suggestions for Authors

Review for the work entitled

The Indirect Cost of E-Mobility: Global Warming Impact of Grid Energy-Mix on E-Mobility Using Real-World Data

The paper may be accepted for publication, but with minimal changes in the following:

1. Check the title. Reword the title, if possible. Think about once more.

2. Inside the abstract section, as an example, instead of, as you wrote, a high-efficiency internal combustion engine vehicle (a Renault Clio), please write: a high-efficiency (please specify diesel or gasoline) engine for the vehicle Renault Clio (specify also MY-Model Year and type, etc.). For further comparison with other research works and results, the subject of the research should be defined as precisely as possible.

3. One should be careful with the term zero emissions. Show if there are possible ways to get electricity. Are wind energy, hydropower or natural gas, etc. used? It is very important to keep in mind that if thermal power plants and coal, if used to produce electricity as primary resources, then they are not ecological. In short, emissions during the production of electricity must also be taken into account when speaking about e-mobility.

In short, if the electricity to drive electric vehicles and charge batteries is obtained from petroleum fuels or from coal, then it is a small benefit in terms of reducing emissions.

I am of the opinion that this fact must be checked and emphasized in every story about electric vehicles.

The facts are that electric vehicles only make sense if the electricity is obtained from renewable energy sources (hydro, wind, thermal sources, etc.)

Another variant is the hybrid drive of modern vehicles with ecologically optimized IC engines.

I am of the opinion that the facts under (iii) should be clearly emphasized, explored, and incorporated within the introductory part.

4. As defined above, it is necessary to expand the introductory part and analyze additional scientific works by researchers from other regions in this field, such as the following:

https://doi.org/10.3390/su11184948

https://doi.org/10.3390/atmos15020184

5. Emphasize the role of hybrid drive as a transition solution to fully electric drive and the importance of economical internal combustion engines in hybrid vehicles. Compare, for example, the European legislation regarding the permitted emission of harmful combustion products.

6. Please state the version number of the used software for linear programming. State the name of the manufacturer, city, and country from where the equipment was sourced.

7. Show the concept of a mobile or home charging station for electric-powered vehicles. Show the network of chargers on highways. Point out problems and needs.

8. The contents of some figures are not legible. Please replace the images with one of a sufficiently high resolution (min. 1000 pixels width/height, or a resolution of 300 dpi or higher).

9. Expand the list of references and the introductory part of the work.

10. For the purpose of comparison, as example, look the difference between zero emission targets and current situation regarding to various electric grid and EV.

11. Check the complete material, especially the terminology, abbreviations, and units of measurement.

12. Expand the discussion of the achieved research results in accordance with the set research goals. Clearly state the contribution of the research.

Author Response

Comment 1:

Check the title. Reword the title, if possible. Think about once more.

Response to Comment 1:

The title has been updated to better reflect the work presented.

Comment 2:

Inside the abstract section, as an example, instead of, as you wrote, a high-efficiency internal combustion engine vehicle (a Renault Clio), please write: a high-efficiency (please specify diesel or gasoline) engine for the vehicle Renault Clio (specify also MY-Model Year and type, etc.). For further comparison with other research works and results, the subject of the research should be defined as precisely as possible.

Response to Comment 2:

The abstract and the main content of the paper have been updated to include the details of the ICE vehicle considered. For the abstract we updated the statement: "the indirect emissions were compared to that of a high-efficiency 1.5l diesel internal combustion engine for the vehicle: 2019 Renault Clio dCi 85."

Comment 3:

"One should be careful with the term zero emissions. Show if there are possible ways to get electricity. Are wind energy, hydropower or natural gas, etc. used? It is very important to keep in mind that if thermal power plants and coal, if used to produce electricity as primary resources, then they are not ecological. In short, emissions during the production of electricity must also be taken into account when speaking about e-mobility.In short, if the electricity to drive electric vehicles and charge batteries is obtained from petroleum fuels or from coal, then it is a small benefit in terms of reducing emissions.

I am of the opinion that this fact must be checked and emphasized in every story about electric vehicles.

The facts are that electric vehicles only make sense if the electricity is obtained from renewable energy sources (hydro, wind, thermal sources, etc.)

Another variant is the hybrid drive of modern vehicles with ecologically optimized IC engines.

I am of the opinion that the facts under (iii) should be clearly emphasized, explored, and incorporated within the introductory part."         

Response to Comment 3:

In fact, it is the objective of this article. The term "zero emissions" is openly used for EVs, mostly for commercial purposes, while only keeping in account the tail to wheel emissions, which in a sense is zero since there is no tail-pipe in an EV. Therefore, in this research work, we took the data of countries having energy mix ranging from low renewable penetration to high renewable penetration to illustrate that the indirect grid carbon emission is in essence part of the ev emissions. In response to the reviewer suggestions, the section is modified as well.

Comment 4:

"As defined above, it is necessary to expand the introductory part and analyze additional scientific works by researchers from other regions in this field, such as the following:https://doi.org/10.3390/su11184948 https://doi.org/10.3390/atmos15020184 "

Response to Comment 4:

The section of literarture review is added and expanded to include more works.

Comment 5:

 Emphasize the role of hybrid drive as a transition solution to fully electric drive and the importance of economical internal combustion engines in hybrid vehicles. Compare, for example, the European legislation regarding the permitted emission of harmful combustion products.

Response to Comment 5:

The scope of this research work is limited to the analysis of electric vehicles vis-a-vis grid energy mix. It is envisaged as a future prospective to work on the impact of hybrid vehicles. The authors are grateful to the reviewer for highlighting the european legislation.

Comment 6:

Please state the version number of the used software for linear programming. State the name of the manufacturer, city, and country from where the equipment was sourced.      

Response to Comment 6:

Version number of Gurobi was added as follows: Both day and annual strategies were modeled as PYOMO [45] concrete models and were solved using the Gurobi solver (version 9.1.2)

Comment 7:

Show the concept of a mobile or home charging station for electric-powered vehicles. Show the network of chargers on highways. Point out problems and needs.       

Response to Comment 7:

The scope of this research work does not include the types of chargers used and the developement of charging infrastructure. It is however limited to evaluating how much carbon emissions is related to the use of  EVs, irrespective of charging infrastructure.

Comment 8:

The contents of some figures are not legible. Please replace the images with one of a sufficiently high resolution (min. 1000 pixels width/height, or a resolution of 300 dpi or higher).

Response to Comment 8:

The Images have been updated with higher resolution images.

Comment 9:

Expand the list of references and the introductory part of the work.

Response to Comment 9:

The authors tried to converge the introduction within the scope of the research work. In case, any pertinent references are missed, the authors humbly request the reviewer to provide the DOI of such references to incorporate in the article.

 Comment 10:

For the purpose of comparison, as example, look the difference between zero emission targets and current situation regarding to various electric grid and EV.

Response to Comment 10:

A table has been added in this regard

Comment 11:

Check the complete material, especially the terminology, abbreviations, and units of measurement.

Response to Comment 11:

We have checked and updated the material with appropriate abbreviations, terminology and units of measurement

Comment 12:

Expand the discussion of the achieved research results in accordance with the set research goals. Clearly state the contribution of the research.        

Response to Comment 12:

The discussion section is modified with respect to well to wheel emissions and tank to wheel emissions to make a good comparison of these two cases. Besides, the optimal charging with respect to the grid energy mix of each country is also discussed.

Reviewer 2 Report

Comments and Suggestions for Authors

The paper provides a meaningful analysis of the indirect emissions of EVs, emphasizing the significance of the energy mix in determining their environmental impact. It offers a balanced view by comparing EVs to high-efficiency internal combustion engine vehicles and highlights areas where improvements can be made. This study is an essential contribution to the ongoing discourse on sustainable transportation and e-mobility transitions, providing practical insights for policymakers and stakeholders in the energy and transportation sectors.

-The figures can be improved to be more visible. 

-The structure of the paper must be improved. 

-The literature section needs to be added. 

_ The methodology section is missing and the chosen method need to be justified. 

Comments on the Quality of English Language

The English quality is good. 

Author Response

Comment 1

The paper provides a meaningful analysis of the indirect emissions of EVs, emphasizing the significance of the energy mix in determining their environmental impact. It offers a balanced view by comparing EVs to high-efficiency internal combustion engine vehicles and highlights areas where improvements can be made. This study is an essential contribution to the ongoing discourse on sustainable transportation and e-mobility transitions, providing practical insights for policymakers and stakeholders in the energy and transportation sectors.              

Response to comment 1

The authors thank the reviewer for his/her compliments.

Comment 2

The figures can be improved to be more visible. 

Response to comment 2

Images have been updated with higher resolution images

Comment 3

The structure of the paper must be improved.    

Response to comment 3:

In accordance with the suggestions of the reviewers, the "introduction" section is split into two parts. The later part is entitled as "Literature review". This is done in order to give more visibility to state of the art and to improve the structure of the article.

Comment 4

The literature section needs to be added.             

Response to comment 4

In accordance with the suggestion of the reviewers, the "literature review" section is added. The authors thank all the reviewers for this suggestion

Comment 5

The methodology section is missing and the chosen method need to be justified. 

Response to comment 5:

The title for section 5 is modified as "Methodology :  Optimal EV charging". The authors thank the reviewer for highlighting this issue.

Reviewer 3 Report

Comments and Suggestions for Authors

Every data table and graph needs to show the source of the data. 

Fig 7 is inaccurate component wise. Please reevaluate.

Eq-1 structurally incorrect.

Should consider level 3 charging as it is the primary means of charging for most vehicles now a days.

Discussion part must include the effect of renewable energy integration and DER generation to mitigate EV vs load problems. 

Comments on the Quality of English Language

OK

Author Response

Comment 1

Every data table and graph needs to show the source of the data. 

Response to Comment 1:

The source of data has been added. The authors thank the reviewer for highlighting this issue.

Comment 2:

Fig 7 is inaccurate component wise. Please reevaluate.    

Response to comment 2:

The figure is not meant to be a complete representation of an EV but rather a reduced  (simplified) representation of the electrical system as it was modeled in the optimizations.

Comment 3:

Eq-1 structurally incorrect.

Response to comment 3

Eq-1 has been updated to be structurally correct

Comment 4:

Should consider level 3 charging as it is the primary means of charging for most vehicles now a days.

Response to Comment 4:

The scope of this research work does not include the types of chargers used and the development of charging infrastructure. It is however limited to evaluating how much carbon emissions is related to the use of EVs. Additionally, because the timestep of the data is 1 hour and the battery capacity is 20kWh, any charging infrastructure beyond 20kW will not yield any benefit with regards to the optimization.

Comment 5

Discussion part must include the effect of renewable energy integration and DER generation to mitigate EV vs load problems. 

Response to comment 5:

Effectively, it is part of the conclusion that more RE and DER penetration in the grid tends to have less indirect CO2 emissions pertained to electric vehicle(s)

Reviewer 4 Report

Comments and Suggestions for Authors

Please take into account the following comments and doubts:
1. 
Please improve the quality of axle descriptions in figures 2, 3, 4, 8 and 11.
2. Please indicate and explain exactly where the data for individual countries in the adopted analyzes were taken from (chapter 3)?
3. There is no description in the text of the data from table 1.
4. On what basis and from where did the authors adopt the data given in Table 1?
5. In line 157, the authors point to Figure 4c. which is not in the text.
6. 
The colors of the data shown in all figures should be the same for individual countries, e.g. red for "DE".
7. The conclusions given in Chapter 7 are too general. This chapter should include more detailed conclusions resulting from the analyses.

Author Response

Comment 1

Please improve the quality of axle descriptions in figures 2, 3, 4, 8 and 11.

Response to Comment 1

Images have been updated with higher resolution images

Comment 2

Please indicate and explain exactly where the data for individual countries in the adopted analyzes were taken from (chapter 3)?   

Response to Comment 2

The appropriate data sources have been added

Comment 3

There is no description in the text of the data from table 1.          

Response to Comment 3

The description is added in the form of this sentence: "For this study, we considered electricity carbon intensity (well-to-wheel) data from Electricitymap [38]. Table 1 shows a summary of the dataset2 used for this study. "

Comment 4

On what basis and from where did the authors adopt the data given in Table 1?  

Response to Comment 4

The data source has been added. The data in the table was calculated based on the the dataset obtained from electricitymap

Comment 5

In line 157, the authors point to Figure 4c. which is not in the text.

Response to Comment 5

The typo error is fixed. In fact, the author wanted to point towards figure 4 (b). The authors thank the reviewer for highlighting this mistake.

Comment 6

The colors of the data shown in all figures should be the same for individual countries, e.g. red for "DE".

Response to Comment 6

Figures have been updated to reflect this change. The authors thank the reviewer for highlighting this issue.

Comment 7

The conclusions given in Chapter 7 are too general. This chapter should include more detailed conclusions resulting from the analyses.              

Response to Comment 7

The conclusion has been modified to reflect the requested changes .

Round 2

Reviewer 4 Report

Comments and Suggestions for Authors

The authors included all comments from the review in the text of the article.